# Population-Specific Differences in Pathogenic Variants of Genes Associated with Monogenic Parkinson’s Disease

**DOI:** 10.3390/genes16040454

**Published:** 2025-04-15

**Authors:** Victor Flores-Ocampo, Amanda Wei-Yin Lim, Natalia S. Ogonowski, Luis M. García-Marín, Jue-Sheng Ong, Dennis Yeow, Claudia Gonzaga-Jauregui, Kishore R. Kumar, Miguel E. Rentería

**Affiliations:** 1Brain & Mental Health Program, QIMR Berghofer Medical Research Institute, Brisbane, QLD 4006, Australia; victor.floresocampo@qimrb.edu.au (V.F.-O.); amanda.lim@qimrb.edu.au (A.W.-Y.L.); natalia.ogonowski@qimrb.edu.au (N.S.O.); luis.garciamarin@qimrb.edu.au (L.M.G.-M.); 2School of Biomedical Sciences, Faculty of Health, Medicine and Behavioural Sciences, The University of Queensland, Brisbane, QLD 4072, Australia; 3Laboratorio Internacional de Investigación sobre el Genoma Humano, Universidad Nacional Autónoma de México, Juriquilla 76230, Querétaro, Mexico; cgonzaga@liigh.unam.mx; 4Population Health Program, QIMR Berghofer Medical Research Institute, Brisbane, QLD 4006, Australia; juesheng.ong@qimrb.edu.au; 5Molecular Medicine Laboratory, Concord Repatriation General Hospital, Concord, NSW 2139, Australia or dennis.yeow1@health.nsw.gov.au (D.Y.); k.kumar@garvan.org.au (K.R.K.); 6Neurology Department, Concord Repatriation General Hospital, Concord, NSW 2139, Australia; 7Sydney Medical School, University of Sydney, Camperdown, NSW 2050, Australia; 8Translational Neurogenomics Group, Genomics and Inherited Disease Program, Garvan Institute of Medical Research, Darlinghurst, NSW 2010, Australia; 9Neuroscience Research Australia, Randwick, NSW 2031, Australia; 10School of Clinical Medicine, UNSW Medicine & Health, St Vincent’s Healthcare Clinical Campus, Faculty of Medicine and Health, UNSW Sydney, Sydney, NSW 2033, Australia; 11School of Biomedical Sciences, Faculty of Health, Queensland University of Technology, Brisbane, QLD 4059, Australia

**Keywords:** familial Parkinson’s disease, monogenic Parkinson, global populations, gnomAD, pathogenic variants

## Abstract

Background: Parkinson’s disease (PD) is a genetically complex neurodegenerative disorder. Up to 15% of cases are considered monogenic. However, research on monogenic PD has largely focused on populations of European ancestry, leaving gaps in our understanding of genetic variability in other populations. This study addresses this gap by analysing the allele frequencies of pathogenic and likely pathogenic variants in known monogenic PD genes across eight global populations, using data from the gnomAD database. Methods: We compiled a list of 27 genes associated with Mendelian PD from the Online Mendelian Inheritance in Man (OMIM) database, and identified pathogenic and likely pathogenic variants using ClinVar. We then performed pairwise comparisons of allele frequencies across populations included in the gnomAD database. Variants with significant frequency differences were further assessed using in silico pathogenicity predictions. Results: We identified 81 variants across 17 genes with statistically significant allele frequency differences between at least two populations. Variants in *GBA1* were the most prevalent among monogenic PD-related genes, followed by *PLA2G6*, *ATP13A2*, *VPS13C*, and *PRKN*. *GBA1* exhibited the greatest variability in allele frequencies, particularly the NM_000157.4:c.1226A>G (p.Asn409Ser) variant. Additionally, we observed significant population-specific differences in PD-related variants, such as the NM_032409.3:c.1040T>C (p.Leu347Pro) variant in *PINK1*, which was most prevalent in East Asian populations. Conclusions: Our findings reveal substantial population-specific differences in the allele frequencies of pathogenic variants linked to monogenic PD, emphasising the need for broader genetic studies beyond European populations. These insights have important implications for PD research, genetic screening, and understanding the pathogenesis of PD in diverse populations.

## 1. Introduction

Parkinson’s disease (PD) is a heterogeneous neurodegenerative disorder with distinct genetic architectures, broadly classified into monogenic (also known as familial or Mendelian PD) and polygenic (also known as idiopathic or complex PD) forms [1]. While the majority of PD cases are polygenic, resulting from the cumulative effects of multiple common variants with small individual effect sizes in combination with environmental factors, approximately 10–15% of cases [2] are attributed to highly penetrant monogenic variants [3,4]. These monogenic forms are characterised by single-gene variants that lead to PD and are typically inherited in either an autosomal dominant or recessive manner. Rare pathogenic variants for monogenic PD are often identified through whole-exome or whole-genome sequencing in affected individuals and families [5,6,7].

To date, more than 20 genes have been implicated in monogenic PD [7,8], with associated phenotypes including idiopathic PD, early-onset PD, and more complex forms of parkinsonism [9,10,11]. Well-established genes linked to autosomal dominant PD include *SNCA*, *LRRK2*, and *VPS35* [12,13]. In contrast, autosomal recessive forms are most commonly associated with variants in *PRKN*, *PINK1*, and *PARK7* (*DJ-1*), which are frequently linked to early-onset disease with a slower disease progression [13,14].

The prevalence of monogenic PD varies across groups, ranging from approximately 5–10% in sporadic cases to 30–40% in early-onset familial PD [2,7,15,16]. In addition, the global distribution of monogenic PD exhibits population-specific variation, with certain variants enriched in specific ancestral groups [17,18]. Differences in variant spectrum and frequency are influenced by complex population dynamics, including genetic drift, founder effects, natural selection, migration, and admixture [19]. Consequently, variants that are rare (i.e., minor allele frequency [MAF] < 0.01) in one population may be more common in another, contributing to differences in PD risk and prevalence worldwide [20].

Understanding population-specific differences in the frequency of pathogenic variants is crucial for developing targeted genetic screening strategies and advancing precision medicine approaches. However, existing PD genetic panels predominantly feature variants common in European ancestry populations, often overlooking clinically relevant understudied variants in other ancestries. This bias contributes to healthcare disparities and hinders the equitable implementation of precision medicine.

In this study, we analysed allele frequencies of pathogenic and likely pathogenic variants in monogenic PD-associated genes in both European and non-European populations, using data from the Genome Aggregation Database (gnomAD) v4.1.0 [21]. gnomAD is a publicly available resource that aggregates and harmonises exome and genome sequencing data from 730,947 exomes and 76,215 whole genomes from over 800,000 individuals, including ~138,000 individuals from non-European groups. It serves as a valuable resource for studying genetic variation in the general population, as it integrates data from both disease-specific studies and general population cohorts, while excluding individuals with severe paediatric diseases. We aimed to identify population-specific differences in pathogenic variant frequencies and explore their implications for PD. This comprehensive analysis of the global distribution of PD-associated pathogenic variation addresses a critical gap in genetic research, ultimately improving our understanding of PD genetics in diverse populations.

## 2. Materials and Methods

### 2.1. Variant Selection

We investigated 27 genes associated with monogenic (Mendelian) forms of PD from the Online Mendelian Inheritance in Man (OMIM) database [22], specifically those from phenotypic series PS168600, and excluding listed non-protein-coding loci. The 27 selected genes included *ADH1C*, *ATP13A2*, *ATXN2*, *ATXN3*, *ATXN8OS*, *CHCHD2*, *DNAJC6*, *EIF4G1*, *FBXO7*, *GBA1*, *GIGYF2*, *HTRA2*, *LRRK2*, *MAPT*, *PARK7*, *PINK1*, *PLA2G6*, *PRKN*, *PSAP*, *PTPA*, *RAB32*, *SNCA*, *SYNJ1*, *TBP*, *UCHL1*, *VPS13C*, and *VPS35* (Appendix A).

Pathogenic and likely pathogenic variants in these genes were queried from the ClinVar online browser (consulted in November 2024) [23], and a table of genomic coordinates was constructed for each variant. To determine reference and alternative alleles, we parsed the canonical Sequence Position Deletion Insertion (SPDI) notation for each variant. In addition, we included pathogenic variants from OMIM not reported in ClinVar, in order to expand the reach of the study. These variants were mapped via rsIDs, which were queried in the Kaviar database [24] to obtain genomic positions and major and minor allele information. Subsequently, for these variants, RefSeq names were obtained from dbSNP.

### 2.2. Testing for Allele Frequency Differences Between Populations

Allele frequencies for ClinVar variants were obtained from gnomAD v4.1.0 online publicly available joint frequency VCF files [21]. gnomAD is a publicly available resource that aggregates and harmonises exome and genome sequencing data from 730,947 exomes and 76,215 whole genomes from over 800,000 individuals, including ~138,000 individuals from non-European groups. It serves as a valuable resource for studying genetic variation in the general population, as it integrates data from both disease-specific studies and general population cohorts, while excluding individuals with severe paediatric diseases. Minor allele counts were extracted from Whole Genome Sequencing and Whole Exome Sequencing data (‘Joint’ dataset) across eight populations, using bcftools 1.9 [25]: African/African American (AFR, n = 37,545), Admixed American (AMR, n = 30,019), Ashkenazi Jewish (ASJ, n = 14,804), East Asian (EAS, n = 22,448), Finnish (FIN, n = 32,026), Middle Eastern (MID, n = 3031), South Asian (SAS, n = 45,546), and Non-Finnish European (NFE, n = 590,031). The Amish population (AMI) was excluded from further analysis due to its small sample size (n < 456). In order to account for truly pathogenic/likely pathogenic variants in the OMIM dataset, only rare variants (MAF < 0.01) were retained.

To assess differences in allele frequencies, we performed Fisher’s exact tests for all possible pairwise comparisons between populations, generating contingency tables for each variant in each comparison using minor and major allele counts from the populations being compared. Major allele counts were calculated by subtracting the minor allele count from the ‘Allele Number (AN)’ in gnomAD’s VCF files, which represents the total number of alleles analysed for a given SNP in a population. To ensure meaningful comparisons, we excluded cases where a population had a minor allele count of zero or where allele count data were unavailable.

To account for false positives, we applied false discovery rate (FDR) correction using the Benjamini–Hochberg procedure. Considering that pairwise population comparisons are independent of one another, we performed FDR correction separately for each set of comparisons. However, within each pairwise comparison set, variants were grouped by gene, as variants in the same gene are not entirely independent, due to linkage disequilibrium. Thus, FDR correction was applied to adjust the *p*-values of all variants in every gene, independently across pairwise comparison sets. Only variants with an FDR-adjusted *p*-value < 0.05 were considered statistically significant (Appendix A)

### 2.3. In Silico Prediction of Variant Pathogenicity

To explore the pathogenicity of the variants in genes associated with monogenic PD and provide a useful reference of in silico pathogenicity assessment for the variants described in this project, we performed variant annotation using ANNOVAR (downloaded Feb 2025) [26] to obtain functional predictions and pathogenicity scores. Variants with statistically significant allele frequency differences in at least one pairwise comparison among the eight ancestral populations were formatted as a variant call format file and were annotated using multiple databases, including ljb26_all (CADD, FATHMM, MetaSVM, MetaLR), refGene (gene-based annotations), and dbNSFP v4.7a (functional predictions for non-synonymous variants). When annotation results overlapped between ljb26_all and dbNSFP v4.7a, we prioritised dbNSFP v4.7a, as it is a more up-to-date database.

We reported the rank scores from the Combined Annotation Dependent Depletion (CADD), Functional Analysis through Hidden Markov Models (FATHMM), MetaSVM, MetaLR, Sorting Intolerant From Tolerant (SIFT), Polymorphism Phenotyping v2 (Polyphen-2), Likelihood Ratio Test (LRT), and Rare Exome Variant Ensemble Learner (REVEL). For comparison purposes, we applied a cutoff of 15 [27] for the CADD phred-scaled scores, classifying variants with scores of ≥15 as ‘deleterious (D)’ and those with scores <15 as ‘tolerated (T)’. This threshold corresponds to the top 5% of most deleterious variants in the human genome. For the REVEL rank scores, we applied a cutoff of 0.75 [28] to classify variants with scores of ≥0.75 as D and those with scores <0.75 as T. For the remaining scores (FATHMM-MKL, FATHMM-XF, MetaSVM, MetaLR, SIFT, Polyphen-2, LRT), we followed the prediction categories provided in the annotation output. These pathogenicity rank scores range from 0 to 1, with values closer to 1 indicating a higher likelihood of the variant being pathogenic. Among the different FATHMM rank scores (FATHMM-MKL, FATHMM-XF), we prioritised FATHMM-MKL due to fewer missing values. Similarly, we prioritised SIFT4G over SIFT because this version of SIFT is optimised for genomic data [29].

## 3. Results

### 3.1. Pathogenic Variant Analysis

We identified 1849 pathogenic/likely pathogenic variants, including single-nucleotide polymorphisms (SNPs), short indels, and structural variants, from ClinVar and OMIM. Of these, only 170 variants were mapped using Kaviar from OMIM, with 48 overlapping with variants already retrieved from ClinVar. A total of 368 of the variants were present in the gnomAD v4.1.0 joint dataset across populations and mapped to 21 of the 27 genes associated with monogenic PD; here, we list these genes in descending order, based on the number of variants identified in ClinVar: *GBA1* (89), *PLA2G6* (73), *ATP13A2* (27), *VPS13C* (24), *PRKN* (23), *PSAP* (23), *PINK1* (19), *SYNJ1* (19), *MAPT* (16), *FBXO7* (14), *GIGYF2* (9), *LRRK2* (7), *HTRA2* (6), *PARK7* (6), *UCHL1* (4), *DNAJC6* (3), *EIF4G1* (2), *ATXN2* (1), *CHCHD2* (1), *PTPA* (1), *SNCA (1).* (Appendix A).

We identified population-specific variants, including nine in AFR, ten in AMR, six in EAS, three in FIN, two in MID, and fourteen in SAS. In addition, five variants were exclusive to the ‘Remaining’ gnomAD classification. Notably, the NFE population exhibited 106 exclusive variants. Variants exclusive to one population were excluded from downstream inter-population comparisons (Figure 1). 

### 3.2. Pairwise Testing of Allele Frequency Differences Across Eight Populations

We conducted 28 pairwise comparisons of allele frequencies across eight populations for each gene in the gnomAD dataset. After applying FDR correction, we identified 81 variants with a statistically significant allele frequency difference in at least one pairwise comparison. These variants were distributed across 17 genes: *GBA1* (22), *PLA2G6* (12), *PINK1* (8), *PRKN* (8), *ATP13A2* (4), *LRRK2* (4), *FBXO7* (3), *GIGYF2* (3), *HTRA2* (3), *PARK7* (3), *VPS13C* (3), *EIF4G1* (2), *PSAP* (2), *ATXN2* (1), *SNCA* (1), *SYNJ1* (1), and *UCHL1* (1). (Appendix A). All reported *p*-values are FDR corrected.

Each variant with a statistically significant allele frequency difference was assigned to the population in which it exhibited the highest frequency. Populations were then ranked by the number of variants they accumulated. The EAS population had the highest number of variants (15 variants across six genes), followed by MID (12 variants in seven genes), AMR (12 variants in seven genes), SAS (11 variants in eight genes), AFR (10 variants in six genes), ASJ (8 variants in four genes), NFE (8 variants in five genes), and FIN (2 variants in two genes). Additionally, three variants were more prevalent in the ‘Remaining’ population group than in any of the eight defined populations (Figure 2).

Across all significant variants, allele frequencies ranged from 1.24 × 10^−3^ (NM_013247.5(HTRA2):c.1195G>A (p.Gly399Ser)) to 1.24 × 10^−6^ (NM_000157.3(GBA1):c.751T>C (p.Tyr251His)) worldwide. Significant variants in the top 10th percentile of global allele frequency were identified in the *HTRA2*, *PRKN*, *GBA1*, *LRRK2*, *GIGYF2*, and *EIF4G1* genes. The largest differences in allele frequency between populations (Δ > 0.009) were observed in the *GBA1*, *LRRK2*, and *HTRA2* genes. The *GBA1* variant NM_000157.4(GBA1):c.1226A>G (p.Asn409Ser) showed allele frequency differences ranging from 0.026 to 0.028 when comparing ASJ against SAS, AFR, MID, FIN, AMR, and NFE populations (*p* < 0.0009). Similarly, the *LRRK2* variant NM_198578.4:c.7153G>A (p.Gly2385Arg) displayed allele frequency differences of 0.025 to 0.026 when comparing EAS to NFE, MID, and SAS groups (*p* < 5.29 × 10^−44^). The *HTRA2* variant NM_013247.5:c.1195G>A (p.Gly399Ser) exhibited differences in allele frequency between 0.0005 and 0.014 (*p* < 0.0001), with SAS exhibiting the greatest divergence compared to EAS, FIN, AFR, AMR, ASJ, and NFE groups. Other genes showing inter-population frequency differences included *PRKN*, *ATP13A2*, and *PARK7* (Figure 3) (Appendix A).

### 3.3. In Silico Pathogenicity Predictions

Of the 81 variants with a statistically significant allele frequency difference in at least one pairwise ancestry comparison, three were short indels and did not have available pathogenicity scores, leaving 78 single-nucleotide variants (SNVs) for in silico prediction (Appendix A). The classification of these variants varied across different prediction tools, with the proportion of variants identified as deleterious ranging from 47% to 97%. The predictions most closely aligned with ClinVar and OMIM pathogenic/likely pathogenic classifications were from CADD and FATHMM-MKL, which classified 76 (97%) and 74 (95%) of the variants as deleterious, respectively. In contrast, REVEL (46, 77%), MetaSVM (44, 72%), MetaLR (43, 71%), LRT (49, 70%), and FATHMM-XF (48, 65%) identified fewer variants as deleterious. SIFT and FATHMM-XF classified 37 (62%) and 34 (57%) of the variants as deleterious, respectively, while PolyphenHVAR classified the lowest proportion, with only 47% of the variants predicted as deleterious.

Out of the 82 variants, 4 were located near a splice site, while the majority were in the exonic region (Appendix A). Among the 78 exonic variants, 58 (74%) were non-synonymous SNVs, and 14 (18%) were stop-gain variants, followed by 3 (4%) start-loss variants, 1 (1%) frameshift deletion, 1 (1%) frameshift insertion, and 1 (1%) non-frameshift deletion. We present the population-specific ancestry findings below.

### 3.4. African (AFR)

We identified 94 pathogenic/likely pathogenic variants reported in ClinVar and OMIM within the AFR population across 17 genes. The highest frequency variant was NM_000157.3(GBA1):c.1444G>A (p.Asp482Asn) (AF = 0.008). Nine variants in *GBA1*, *PLA2G6*, *VPS13C*, and *GIGYF2* were exclusive to this population, with allele frequencies ranging from 1.33 × 10^−5^ to 9.38 × 10^−5^. Comparing AFR to other populations, 39 pathogenic variants showed significant allele frequency differences across the 17 genes, except for *ATXN2*, *PSAP*, and *UCHL1*. Of these 39 pathogenic variants, 10 were the most prevalent in AFR, occurring in *GBA1*, *ATP3A2*, *PARK7*, *VPS13C*, *SYNJ1*, and *PLA2G6*, with significant frequency differences compared to AMR, MID, NFE, and SAS populations. Finally, the largest allele frequency difference was for NM_000157.4(GBA1):c.1226A>G (p.Asn409Ser) (AFR *=* 0.0002 vs. ASJ = 0.028).

### 3.5. Admixed American (AMR)

For the AMR population, we observed 92 pathogenic/likely pathogenic variants across 14 genes. In this population, the pathogenic variant NM_004562.3:c.823C>T (p.Arg275Trp) in *PRKN* was the most common, with an allele frequency of 1.85 × 10^−3^. Furthermore, ten variants specific for this population were detected across the genes *GBA1*, *PLA2G6*, *VPS13C*, *PRKN*, and *ATP13A2*, with allele frequencies ranging from 1.72957 × 10^−4^ to 1.67 × 10^−5^. In comparison to other populations, 36 pathogenic variants exhibited significant differences in allele frequency across 17 genes, except *VPS13C*, *SYNJ1*, *PLA2G6*, *UCHL1*, and *SNCA*. Out of these pathogenic variants, 12 were most prevalent in AMR, occurring in *GBA1*, *ATP3A2*, *PSAP*, *ATXN2*, *LRRK2*, *FBXO7*, and *PRKN*, with significant frequency differences compared to NFE, FIN, and SAS populations. The most common pathogenic variant with a statistically significant frequency difference was NM_002778.4:c.650C>T (p.Thr217Ile) in *PSAP* (AMR *=* 3.50 × 10^−4^, for reference global = 1.36 × 10^−5^). The largest allele frequency difference between AMR and any other population was observed for the variant NM_000157.4:c.1226A>G (p.Asn409Ser) in *GBA1* (*p*~0), with a frequency of 9.50 × 10^−4^ in AMR, compared to 0.028 in ASJ.

### 3.6. Ashkenazi Jewish (ASJ)

We identified 18 pathogenic/likely pathogenic variants for the ASJ population, involving seven genes. However, no pathogenic variants were exclusively found in ASJ. Comparing ASJ to other populations, 12 pathogenic variants showed significant differences in allele frequency across *GBA1*, *ATP13A2*, *LRRK2*, *GIGYF2*, *HTRA2*, *AMD*, and *PRKN*. Eight pathogenic variants were the most prevalent in ASJ, occurring in *GBA1*, *ATP3A2*, *LRRK2*, and *PRKN*. In addition, these variants showed significant frequency differences when compared to all other populations.

The most common pathogenic variant with a significant frequency difference was NM_000157.4(GBA1):c.1226A>G (p.Asn409Ser) (ASJ *=* 0.028, for reference global = 0.002). Furthermore, the same variant represented the largest allele frequency difference between any pairwise comparison in the whole dataset when comparing ASJ against SAS, AFR, MID, FIN, AMR, and NFE (*p* < 3.62 × 10^−60^).

### 3.7. East Asian (EAS)

In the EAS population, we uncovered 69 variants spanning across 15 genes. Of these 69 pathogenic variants, 6 were exclusively found in EAS, with allele frequencies ranging from 2.23 × 10^−4^ to 2.23 × 10^−5^, in the genes *PLA2G6*, *PRKN*, *VPS13C*, *GBA1*, and *PARK7*. Comparing EAS to other populations, 26 pathogenic variants showed significant differences in allele frequency across *GBA1*, *PINK1*, *LRRK2*, *GIGYF2*, *HTRA2*, *PLA2G6*, and *PRKN*. Moreover, 15 variants were the most prevalent in EAS, occurring in *GBA1*, *PINK1*, *LRRK2*, *GIGFY2*, *HTRA2*, and *PLA2G6*. These 15 variants showed significant frequency differences when compared to AFR, AMR, FIN, MID, NFE, and SAS.

The most common pathogenic variant was NM_198578.4:c.7153G>A (p.Gly2385Arg) in *LRRK2* (EAS = 0.026, for reference global = 7.87 × 10^−4^). The largest allele frequency difference between EAS and any other population was observed for the same variant, NM_198578.4(LRRK2):c.7153G>A (p.Gly2385Arg), when comparing it against NFE (AF = 1.10 × 10^−5^, *p*~0), MID (AF = 1.65 × 10^−4^, *p* = 1.86 × 10^−63^), and SAS (AF = 5.38 × 10^−4^, *p*~0).

### 3.8. Finnish (FIN)

For the FIN population, 33 pathogenic/likely pathogenic variants in 11 genes were identified. The predominant variant was NM_004562.3:c.823C>T (p.Arg275Trp) in *PRKN*, which also represented the most common pathogenic variant in the FIN population, exhibiting an allele frequency of 9.84 × 10^−4^. Three variants were exclusively observed among the FIN population, with allele frequencies ranging from 3.15 × 10^−5^ to 1.56 × 10^−5^ across *ATP13A2*, *GBA1*, and *GIGYF2*.

Comparing FIN to other populations, 14 pathogenic variants showed significant differences in allele frequency across *VGBA1*, *ATP13A2*, *GBA1*, *ATXN2*, *VPS13C*, *GIGYF2*, *LRRK2*, *HTRA2*, *GBA1*, *PLA2G6*, and *PRKN*. Of these 14 pathogenic variants, 2 were the most prevalent for FIN, namely, NM_022089.4(ATP13A2):c.1657C>T (p.Arg553Ter) and NM_020821.3(VPS13C):c.9019C>T (p.Arg3007Ter). Notably, these variants showed significant frequency differences when compared primarily to NFE (*p* = 0.02 and *p* = 7.18 × 10^−13^, respectively).

The pathogenic variant (p.Arg3007Ter) in *VPS13C* showed a frequency of 2.03 × 10^−4^, compared to 1.30 × 10^−5^ worldwide. Consequently, it was the most prevalent pathogenic variant, exhibiting a notable frequency difference within the FIN population. The largest allele frequency difference for FIN was observed for NM_000157.4(GBA1):c.1226A>G (p.Asn409Ser) (FIN = 9.06 × 10^−4^ vs ASJ = 0.028, *p*~0).

### 3.9. Middle Eastern (MID)

We identified 29 pathogenic/likely pathogenic variants for the MID population, mapping to 11 genes. The variant NM_013247.5:c.1195G>A (p.Gly399Ser) in HTRA2 was the most common variant (AF = 0.009). In addition, two variants were exclusively identified in MID, namely NM_000157.4:c.518C>A (p.Thr173Asn) in *GBA1* (AF = 4.95 × 10^−4^) and NM_002778.4:c.778-1889C>T (AF = 2.12 × 10^−4^) in *PSAP*.

We observed 17 pathogenic variants with significant differences in allele frequency across all of *GBA1*, *PINK1*, *PARK7*, *LRRK2*, *VS13C*, *GIGYF2*, *HTRA2*, *PLA2G6*, and *PRKN*. Of these pathogenic variants, eight were the most prevalent in MID, occurring in *PINK1*, *PARK7*, *LRRK2*, *VPS13C*, *PLA2G6*, and *PRKN*. These variants showed significant frequency differences compared to AFR, AMR, SAS, and NFE.

The variant NM_198578.4:c.4321C>T (p.Arg1441Cys) in *LRRK2* (AF = 0.001) was the most common pathogenic variant with a significant frequency difference in MID comparisons, with a frequency of 0.001, compared to 2.42 × 10^−5^ worldwide. The largest allele frequency difference was for NM_198578.4:c.7153G>A (p.Gly2385Arg) in *LRRK2* (MID = 1.65 × 10^−4^ vs EAS = 0.026, *p* = 1.86 × 10^−63^).

### 3.10. Non-Finnish European (NFE)

For the NFE population, 287 pathogenic/likely pathogenic variants spanning over 20 genes were identified. The most common variant in NFE was NM_004562.3:c.823C>T (p.Arg275Trp) in *PRKN* (AF = 3.73 × 10^−3^). Out of the 287 pathogenic variants, we observed 106 to be exclusively found for NFE, with allele frequencies ranging from 2.89 × 10^−5^ to 8.47 × 10^−7^, for the genes *GBA1*, *PLA2G6*, *PINK1*, *PRKN*, *ATP13A2*, *LRRK2*, *FBXO7*, *GIGYF2*, *HTRA2*, *PARK7*, *VPS13C*, *EIF4G1*, *PSAP*, *ATXN2*, *SNCA*, *SYNJ1*, and *UCHL1*.

Comparing NFE to other populations, 77 pathogenic variants showed significant differences in allele frequency across all significant genes. Of these variants, eight were most prevalent in NFE, occurring in *GBA1*, *GIGYF2*, *EIF4G1*, *SNCA*, and *PRKN*. These variants also showed significant frequency differences when compared to all other populations.

The most common pathogenic variant with a significant frequency difference was NM_004562.3:c.823C>T (p.Arg275Trp) in *PRKN* (NFE *=* 0.004, for reference global = 0.003), while the largest allele frequency difference was for NM_000157.4:c.1226A>G (p.Asn409Ser) in *GBA1* (NFE = 0.002 vs. ASJ = 0.028, *p*~0).

### 3.11. South Asian (SAS)

Lastly, 99 pathogenic/likely pathogenic variants, mapping to 17 genes, were reported for SAS. The most common pathogenic variant was NM_013247.5:c.1195G>A (p.Gly399Ser) in *HTRA2* (AF = 1.44 × 10^−2^). Furthermore, 14 variants were exclusively identified in the SAS population, with allele frequencies ranging from 1.29 × 10^−4^ to 1.10 × 10^−5^, across *PLA2G6*, *GBA1*, *PINK1*, *PSAP*, *PRKN*, *FBXO7*, and *ATP13A2*. Comparing SAS to other populations, 38 pathogenic variants showed significant differences in allele frequency across all 17 genes, except for *ATP13A2*, *VPS13C*, *GIGYF2*, and *EIF4G1*. Of these, 11 were most prevalent in SAS, occurring in *PINK1*, *PRKN*, *PARK7*, *PSAP*, *HTRA2*, *FBXO7*, *PLA2G6*, and *UCHL1*. These variants showed significant frequency differences when compared to all the other population groups.

The most common pathogenic variant with a statistically significant frequency difference was p.Gly399Ser in *HTRA2* (SAS = 0.014, for reference global = 0.004), which was classified as deleterious (CADD Phred-scaled rank score = 27.2, FATHMM-MKL rank score = 0.596, FATHMM-XF rank score = 0.793, MetaSVM rank score = 0.911, MetaLR rank score = 0.970). The results revealed that the difference in allele frequency between SAS (AF = 5.49 × 10^−5^) and ASJ (AF = 0.028) for the NM_000157.4(GBA1):c.1226A>G (p.Asn409Ser) was the largest in the whole dataset (*p*~0).

## 4. Discussion

The present study represents a comprehensive investigation into the genetic diversity of monogenic PD across global populations, a critical gap in genetic research. By analysing allele frequencies of pathogenic and likely pathogenic variants in known Mendelian PD genes using gnomAD data, we provide insights into inter-population genetic variability. We identified 1849 pathogenic or likely pathogenic variants across 27 PD-associated genes, with *GBA1*, *PLA2G6*, *ATP13A2*, *VPS13C*, and *PRKN* being the most frequently represented. Among these, 368 variants were present in gnomAD across all populations, emphasising the need for population-specific genetic screening approaches. In addition, our results revealed 81 variants across 17 genes, with statistically significant differences in allele frequency between at least two populations, underscoring the substantial genetic heterogeneity in PD. The EAS population exhibited the highest number of differentiated variants (15 across six genes), followed by MID (12), AMR (12), and SAS (11).

Population-specific analyses further emphasised the distinct genetic architectures of monogenic PD forms across diverse populations. Pathogenic variants in the *GBA1* gene, which encodes the enzyme β-glucocerebrosidase, are well-established risk factors for α-synucleinopathies, including PD [30]; however, their impact on the clinical spectrum of PD remains unclear [30]. Among the pathogenic variants examined, *GBA1* exhibited the largest variability in allele frequencies, particularly for the p.Asn409Ser variant. This variant showed the highest frequency in ASJ compared to other populations, including AFR, SAS, AMR, MID, FIN, and NFE. Our findings are consistent with previous studies that have reported a high frequency of *GBA1* PD pathogenic variants in ASJ PD patients, with 15% of ASJ participants carrying at least one *GBA1* variant, including NM_000157.4:c.1448T>C, known as p.Leu444Pro and p.Asn409Ser [30,31]. Our results reveal significant differences in allele frequency for both pathogenic variants. Notably, p.Leu444Pro showed the highest allele frequency in the MID population, while p.Asn409Ser was most prevalent in ASJ. Similar observations have been made in mainland China, where over 8% of individuals with idiopathic PD carry a *GBA1* variant, with p.Leu444Pro being the most common [32]. The substantial allele frequency difference suggests the influence of population-specific selection or genetic drift, underscoring the need for customised screening strategies for different populations.

*LRRK2* has also been widely characterised for its major involvement in monogenic PD [33]. This gene encodes leucine-rich repeat kinase 2, a protein involved in various cellular processes, including vesicle trafficking and autophagy, both of which are crucial for maintaining neuronal health [34]. The p.Gly2019Ser variant is the most common monogenic cause of PD, accounting for up to 4% of cases worldwide [17,35], with a particularly high prevalence in ASJ and North African populations [35]. Yet, its prevalence varies widely even across broader African populations, with studies reporting an almost complete absence of *LRRK2* pathogenic variants in black Africans and Nigerian cohorts (gnomAD F = 4.00 × 10^−5^) [17,36,37]. Our analysis confirms significant allele frequency variability, with the largest difference observed between ASJ and FIN populations, while SAS showed no reported cases. Furthermore, this heterogeneity extends to AMR, where previous studies within the Latin American Research Consortium on the Genetics of PD (LARGE-PD) reported considerable variability in the prevalence of the *LRRK2* p.Gly2019Ser variant across Latin American countries, with frequencies ranging from 0.002 in Peru to 0.042 in Uruguay [38]. In contrast, our analysis identified that p.Arg1441Gly, commonly observed in European populations, displayed significant allelic frequency differences between NFE and AMR [38], suggesting differential selective pressures or founder effects. Similarly, p.Gly2385Arg, a variant prevalent in EAS, demonstrated the largest allele frequency differences when compared to NFE (AF = 1.10 × 10⁻^5^, *p*~0), and was previously identified in 11.65% of idiopathic PD cases in Japanese cohorts [17,39].

In addition, the p.Arg1441Cys variant represents an alternative allele at the same locus as p.Arg14441Gly, leading to a different residue substitution. This pathogenic variant was reported to have the highest frequency difference in the MID population (0.001) compared to the global frequency (2.42 × 10^−5^). However, the largest allele frequency difference in MID was observed for G2385A in *LRRK2* when compared to EAS (MID = 1.65 × 10⁻^4^ vs. EAS = 0.026, *p* = 1.86 × 10^−63^). Beyond GBA1 and LRRK2, other candidate genes have been implicated in monogenic PD forms, though their roles remain controversial. One such gene is *HTRA2*, where ancestry-specific investigations have yielded conflicting results regarding its contribution to monogenic PD. This gene encodes a mitochondrial serine protease involved in cellular stress response and mitochondrial maintenance [40]. Recent studies in Chinese [41] and Taiwanese [42,43] populations have identified potentially pathogenic *HTRA2* variants, whereas large-scale analyses in European, Ashkenazi Jewish [44], and Latin American [45] cohorts found no significant association with PD.

Three missense variants in *HTRA2*—p.Pro143Ala, p.Gly399Ser, and p.Arg404Gln—exhibited significant allele frequency differences across ancestries. The p.Pro143Ala variant was most prevalent in the EAS population (AF = 8.92 × 10^−5^), with significantly higher allele frequency than in the NFE population (AF = 8.48 × 10^−7^, *p* = 1.30 × 10⁻^5^). Functional studies support a pathogenic role, as p.Pro143Ala expression in dopaminergic neurons induces neurite degeneration [43]. Additionally, case–control studies in Taiwanese cohorts [42,43] have reported this variant in both early- and late-onset PD cases, but not in controls, suggesting a possible association with PD risk. On the other hand, the p.Gly399Ser variant is the most common pathogenic *HTRA2* variant in SAS (AF = 0.014) and MID populations (AF = 8.63 × 10^−3^). Although in silico predictions suggest a deleterious effect, no experimental studies currently support its pathogenicity. Similarly, p.Arg404Gln, which is most prevalent in SAS (AF = 3.29 × 10^−5^) and significantly more frequent in NFE (AF = 7.63 × 10^−6^, *p* < 0.049), lacks experimental validation. The absence of experimental evidence for these two variants underscores the importance of population-specific studies to assess their clinical relevance in PD.

Our findings reinforce the role of *PRKN*, which encodes the E3 ubiquitin protein ligase parkin, as a predominant genetic contributor to early-onset PD [46]. *PRKN*-associated PD is characterised by a distinct and severe loss of dopaminergic neurons of the substantia nigra pars compacta, which differentiates it from the neurodegenerative patterns observed in idiopathic PD [47]. Consistent with prior studies [6,45,46], we observed notable *PRKN* variant allele frequencies in AMR and NFE populations, with the pathogenic variant p.Arg275Trp being most prevalent (AF = 9.84 × 10⁻^4^ in FIN, 3.73 × 10⁻^3^ in NFE) These findings support *PRKN* as the most common autosomal recessive juvenile PD gene, accounting for ~27.6% of cases in European cases [6,46].

Additionally, our analysis revealed significant allele frequency differences between AMR and both NFE and EAS populations. In contrast to these findings, none of the *PRKN* variants identified in previous studies were significantly variable in EAS populations, highlighting the need for further investigation. Although our results support the role of *PRKN* in AMR PD genetics, we identified two variants, p.Met1Valand and p.Cys212Tyr, with the highest allele frequencies in the AMR population, neither of which overlapped with previously reported variants. On the other hand, previous studies in Indian cohorts have reported that *PRKN* carries point variants in ~7.7% of patients, accounting for 16.65% of familial PD cases in these populations [48]. While the specific variants identified in this study were not included in our analysis, we reported a significant variant that is most prevalent in the SAS population. The variant p.Met1Thr, with a MAF of 0.000460289, is a start-loss variant located in an exonic region.

Beyond *PRKN*, *PINK1* and *PARK7 (DJ-1)* have been recognised as additional contributors to autosomal recessive PD. *PINK1*, which encodes PTEN-induced kinase 1, plays a crucial role in mitochondrial quality control, particularly through its involvement in the mitochondrial damage response and mitophagy [49]. Previous studies have indicated that *PINK1* is the second most relevant cause of PD, particularly in populations such as EAS and Indian populations, where 0.072 to 0.25 of patients with early-onset PD carry pathogenic variants in this gene [6,17,50]. In particular, the *PINK1* variant NM_032409.3:c.1040T>C (p.Leu347Pro) has been established as a predominant pathogenic allele in EAS, though absent in AFR, ASJ, FIN, MID, and SAS populations.

Similarly, at least 20 variants in *PARK7 (DJ-1)*, which encodes a protein involved in oxidative stress response and mitochondrial maintenance, have been implicated in monogenic PD [50]. *PARK7* has been associated with more severe PD phenotypes, with pathogenic variants observed in up to 0.039 of PD patients in eastern India [51]. In our study, out of the three significant pathogenic variants in *PARK7*, p.Arg28Gln was the most common in SAS (AF = 3.84 × 10^−4^). Notably, this variant has not been characterised in prior studies, suggesting that it may represent a novel genetic contributor to PD in this population.

In addition to these well-established monogenic PD genes, our study also examined genetic variation in genes associated with complex or atypical Mendelian forms of PD. Although pathogenic variants in these genes are less frequent, their variability across populations highlights the need for further investigation to better understand their contribution to PD. For instance, *PSAP* has been implicated in PD, with growing evidence linking its variants to sporadic, familial, and early-onset forms of the disease [52,53]. Recent studies have reported exonic and intronic *PSAP* variants associated with an increased risk of PD monogenic forms, particularly in Asian populations [54]. In our study, we identified p.Thr217Ile as the most prevalent pathogenic variant in the AMR population, while the pathogenic variant p.Gln260Ter was more common in the MID population. Additionally, several *PSAP* pathogenic variants were observed exclusively in NFE and SAS populations, further demonstrating population-specific genetic contributions to PD.

On the other hand, we explored pathogenic variants in *VPS13C*, an autosomal recessive early-onset PD gene. *VPS13C* encodes a protein involved in vesicular trafficking and lipid homeostasis, crucial for maintaining cellular function, especially in neurons. Biallelic loss-of-functions *VPS13C* variants have been reported in monogenic PD, while heterozygous variants may contribute to complex PD susceptibility [50,55]. Our analysis identified p.Arg3007Ter as the most prevalent pathogenic variant in the FIN population, with an allele frequency of 2.03 × 10⁻^4^, significantly higher than the 1.30 × 10⁻^5^ observed globally, suggesting potential founder effects or genetic drift in *VPS13C*-related PD risk in FIN. We also observed exclusive *VPS13C* pathogenic variants in the NFE population. Despite an increasing incidence of early-onset PD in Finland [56], studies analysing monogenic causes remain limited. Genome-wide analyses of Finnish PD cohorts suggest that no single monogenic cause predominates in this population. This aligns with our findings that VPS13C variants show moderate allele frequencies without a clear monogenic pattern [57]. Additionally, we identified the p.Arg553Ter variant in *ATP13A2*, another lysosomal-related PD gene, as significantly enriched in FIN. *ATP13A2* encodes a protein involved in maintaining lysosomal integrity and cellular homeostasis, which is relevant for the degradation of neurotoxic proteins [58]. Lastly, *SNCA*—which encodes α-synuclein, a protein central to the pathogenesis of synucleinopathies—has been characterised as a relatively rare PD causal gene in non-European populations [7]. We were able to identify a single-nucleotide variant, p.His50Gln, with significantly different allele frequencies between NFE (AF = 2.33 × 10^−4^), SAS (1.10 × 10^−5^, *p* = 5.89 × 10^−8^), and AFR (4.01 × 10^−5^, *p* = 6.84 × 10^−5^) populations.

Despite the NFE population having the largest count for pathogenic variants in PD-related genes, among variants with significant allele frequency differences, only eight variants were most common in this group. Arguably, the high number of identified pathogenic variants is related to the exhaustive research performed on this group, and is not directly related to a higher incidence of monogenic PD. Indeed, the current understanding of PD-associated genetic variants is limited by a prominent Eurocentric bias, as most genetic studies have focused predominantly on populations of European ancestry. This bias creates a critical gap in our advancement of the genetics of PD in non-European populations, ultimately hindering the efforts to address global health disparities in PD diagnosis and treatment.

While some of the differences in PD prevalence reported in the literature may reflect true biological variation, this may not be the only factor determining the prevalence of monogenic PD across populations. In fact, the prevalence could also be influenced by differences in the degree of population ageing, as well as disparities in healthcare access, diagnostic accuracy, and awareness of the disease across geographical regions [59]. Consequently, PD might be underdiagnosed or under-reported in countries with a lower human development index or socio-demographic index [60,61].

To date, this study represents the most comprehensive analysis of pathogenic variants in PD genes across globally diverse populations. However, certain limitations must be acknowledged. For instance, the allele frequency data from gnomAD may reflect an approximation to the prevalence of PD pathogenic and likely pathogenic variants in the general population; however, gnomAD’s cohort selection framework introduces biases, as its samples are mostly derived from cohort studies focused on specific diseases or traits, which may not accurately represent broader population distributions. Moreover, ultra-rare variants, large indels, or structural variants may be difficult to capture and analyse in this dataset. On the other hand, the penetrance of the variants described in this study varies widely—some act as primary disease-causing variants, while others might function as risk factors for idiopathic PD or other diseases [6]. Given the absence of available phenotypic data for gnomAD, it is not possible to assess the penetrance and expressivity of the identified variants in the carrier individuals.

Our findings pose significant implications for the putative design of targeted screening strategies. In particular, the population-specific differences in the frequencies of pathogenic variants highlight the need to tailor screening strategies based on a given ancestry. For instance, in populations where certain variants (e.g., specific mutations in GBA1 or LRRK2) are more prevalent, incorporating these variants into genetic tests could potentially improve diagnostic tools and risk assessment. This targeted approach would not only enhance early detection, but also represent a step forward in patient-specific clinical genetic counselling and intervention strategies relevant to an individual’s ancestral background. In addition, by understanding the prevalence of pathogenic variants across global populations, we could better stratify patients for gene-targeted therapies. For example, individuals harbouring high-risk variants may benefit from earlier or more aggressive intervention, or may be ideal candidates for enrolment in clinical trials testing variant-specific treatments.

## 5. Conclusions

In conclusion, we comprehensively analysed and described the variability of allele frequencies of pathogenic variants in Mendelian PD-related genes across eight populations, while also characterising their pathogenicity and highlighting those variants of particular importance to specific populations. By documenting population-specific differences in allele frequency, our findings provide a valuable reference for both clinical and research applications, facilitating the identification of potential genetic targets for further study and precision medicine efforts.

## Figures and Tables

**Figure 1 genes-16-00454-f001:**
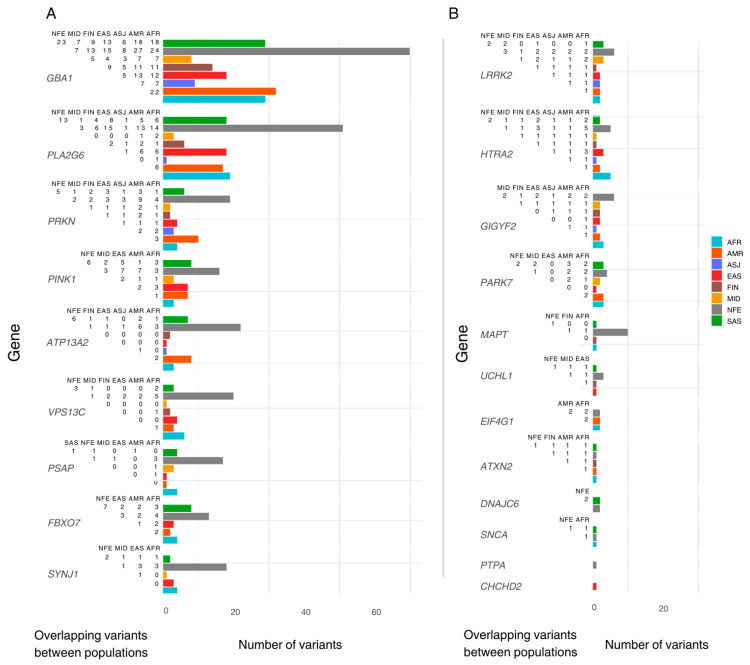
The number of pathogenic variants reported per ancestry and overlap between populations. The *Y*-axis lists genes included in this study, ordered by the total number of pathogenic/likely pathogenic (P/LP) variants reported. The *X*-axis shows the total number of P/LP variants found in ClinVar/OMIM per ancestry, with coloured bars representing variants found in the gnomAD dataset per population. Each population includes both exclusive and shared variants. To the left of each bar, a half-matrix indicates the overlap of variants between populations. Each number in the matrix represents the count of overlapping variants between the population associated with the corresponding coloured bar and the population listed in each column. (**A**) lists genes with more than 18 variants reported (population with highest count). (**B**) lists the genes with less than 18 variants reported (population with highest count). AFR: African/African American; AMR: Admixed American; ASJ: Ashkenazi Jewish; EAS: East Asian; FIN: Finnish; MID: Middle Eastern; SAS: South Asian; NFE: Non-Finnish European.

**Figure 2 genes-16-00454-f002:**
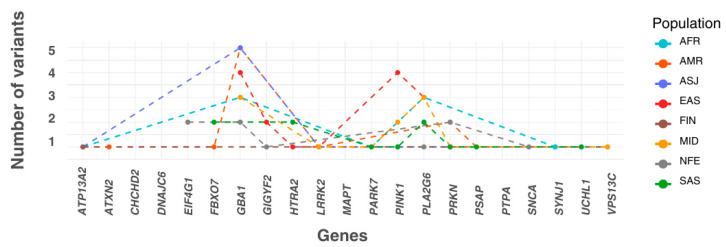
The numbers of most frequent significant pathogenic variants across ancestry. The *X*-axis lists the genes included in this study in alphabetical order. The plot shows the number of significant variants assigned to each population, where each variant is categorised based on the population in which it has the highest allele frequency. In this panel, the *Y*-axis represents the number of variants accumulated by each population for every gene. AFR: African/African American; AMR: Admixed American; ASJ: Ashkenazi Jewish; EAS: East Asian; FIN: Finnish; MID: Middle Eastern; SAS: South Asian; NFE: Non-Finnish European.

**Figure 3 genes-16-00454-f003:**
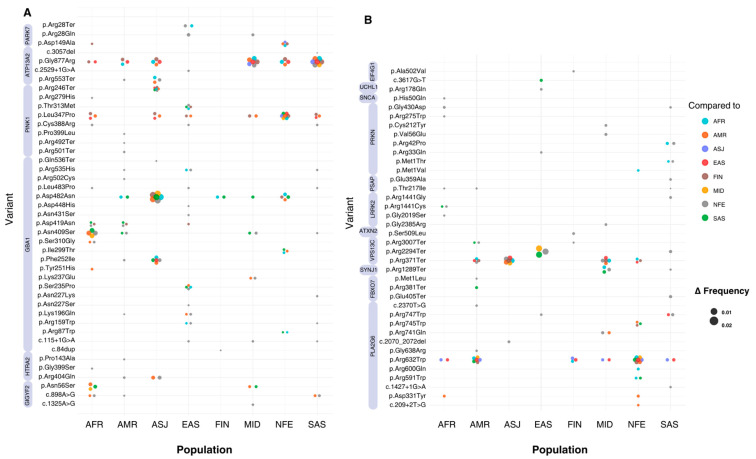
Absolute allele frequency differences between populations for variants with significant frequency differences. The *X*-axis represents the eight populations included in this study, while the *Y*-axis lists all the variants that show significant differences in allele frequency. Bubbles at variant-population intersections indicate a statistically significant difference between the population on the *X*-axis and another population, as specified by the bubble’s colour. The population on the *X*-axis represents the group with the highest allele frequency for each variant. The absolute allele frequency difference (Δ) is represented by bubble size, with larger bubbles indicating greater Δ values. Variants with multiple significant pairwise differences between populations appear as clusters of bubbles (or ‘flowers’). (**A**) includes all variants located on chromosomes 1 and 2, while (**B**) includes variants from chromosomes 3 to 22. Violet bands next to variant labels indicate the genes to which each variant is mapped.

## Data Availability

We used publicly available data from public databases. Code developed during the project can be obtained from the following repository: https://github.com/computational-neurogenomics/PD_global_Frequencies.git.

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
