# Peer review of "Population-Specific Differences in Pathogenic Variants of Genes Associated with Monogenic Parkinson’s Disease"

_genes, 2025, doi:10.3390/genes16040454_

Round 1

Reviewer 1 Report

Comments and Suggestions for Authors

Manuscript is a comprehensive analysis of pathogenic vari-529 ants in PD genes across globally diverse populations.

The content is written in correct English. The authors performed many analyses using databases and various bioinformatic tools. This allowed them to identify variants whose frequency differs between populations.

The authors did not avoid minor terminological errors concerning the notation of variants (e.g. omitting the letter p before the variant symbol at the protein level).

The content would be more interesting if, when describing the most important identified variants, the authors provided information on what proteins are encoded by these genes and tried to determine what significance they may have for a better understanding of the mechanisms important in the pathogenesis of PD.

In the introduction - justifying the purpose of conducting analyses - the authors indicated that "Understanding population-specific differences in the frequency of pathogenic variants is crucial for developing targeted genetic screening strategies and advancing precision medicine approaches.". However, in the discussion they did not specify how their findings could be relevant to the development of targeted screening strategies or advanced applications of precision medicine - this is worth adding.

The last paragraph of the introduction contains information that concerns the methodology (lines 79-87).

The last paragraph of the Materials and Methods section is a copy of the Genes editors' instructions for preparing manuscripts - it should be removed.

Figure 1: the scale of the Y-axis should be wider, because some values ​​are higher than indicated on the Y-axis (applies to both A and B).

It would be worth adding a table containing data on previously identified and new genes, the variants of which are associated with PD.

Unfortunately, the link provided as a reference to the source of supporting information www.mdpi.com/xxx/s1 does not work - on the page that opens at this address, the message Error 404 - File not found is displayed

Author Response

Reviewer 1

The manuscript is a comprehensive analysis of pathogenic variants in PD genes across globally diverse populations.

The content is written in correct English. The authors performed many analyses using databases and various bioinformatic tools. This allowed them to identify variants whose frequency differs between populations.

We thank the reviewer for taking the time to read and review our study. We have improved the quality of our work based on the comments provided.

  • The authors did not avoid minor terminological errors concerning the notation of variants (e.g. omitting the letter p before the variant symbol at the protein level).

We appreciate the reviewer’s attention to detail regarding variant notation. We have revised the manuscript to ensure that all variant names adhere to the HGVS nomenclature, including the correct use of "p." before protein-level changes.

  • The content would be more interesting if, when describing the most important identified variants, the authors provided information on what proteins are encoded by these genes and tried to determine what significance they may have for a better understanding of the mechanisms important in the pathogenesis of PD.

We thank the reviewer for the suggestion. We have expanded our descriptions of the key proteins encoded by the identified genes and their relevance to the pathogenesis of PD. We have incorporated the following additional information into the Discussion section:

“Pathogenic variants in the GBA1 gene, which encodes the enzyme beta-glucocerebrosidase, are well-established risk factors for alpha-synucleinopathies, including PD [30], however, their impact on the clinical spectrum of PD remains unclear [30]”

“This gene encodes leucine-rich repeat kinase 2, a protein involved in various cellular processes, including vesicle trafficking and autophagy, both of which are crucial for maintaining neuronal health [34].”

“This gene encodes a mitochondrial serine protease involved in cellular stress response and mitochondrial maintenance [40].”

“Our findings support the role of PRKN, which encodes the E3 ubiquitin protein ligase parkin, as a predominant genetic contributor to early-onset PD [43]. PRKN-associated PD is characterised by a distinct and severe loss of dopaminergic neurons of the substantia nigra pars compacta, which differentiates it from the neurodegenerative patterns observed in idiopathic PD [47].”

“PINK1, which encodes PTEN-induced kinase 1, plays a crucial role in mitochondrial quality control, particularly through its involvement in the mitochondrial damage response and mitophagy [49].”

“Similarly, at least 20 variants in PARK7 (DJ-1), which encodes a protein involved in oxidative stress response and mitochondrial maintenance, have been implicated with more severe in monogenic PD form[45].”

VPS13C encodes a protein involved in vesicular trafficking and lipid homeostasis, crucial for maintaining cellular function, especially in neurons.”

ATP13A2 encodes a protein involved in maintaining lysosomal integrity and cellular homeostasis, which is relevant for the degradation of neurotoxic proteins.”

“Lastly, SNCA -which encodes alpha-synuclein, a protein central to the pathogenesis of synucleinopathies-”

  • In the introduction - justifying the purpose of conducting analyses - the authors indicated that "Understanding population-specific differences in the frequency of pathogenic variants is crucial for developing targeted genetic screening strategies and advancing precision medicine approaches.". However, in the discussion they did not specify how their findings could be relevant to the development of targeted screening strategies or advanced applications of precision medicine - this is worth adding.

We thank the reviewer for this remark. We have added the following text to the discussion:

“Our findings pose significant implications for the putative design of targeted screening strategies. In particular, the population-specific differences in the frequencies of pathogenic variants highlight the need to tailor screening strategies based on a given ancestry. For instance, in populations where certain variants (e.g., specific mutations in GBA1 or LRRK2) are more prevalent, incorporating these variants into genetic tests could potentially improve diagnostic tools and risk assessment. This targeted approach would not only enhance early detection but also represent a step forward in patient-specific clinical genetic counselling and intervention strategies relevant to an individual's ancestral background. In addition, by understanding the prevalence of pathogenic variants across global populations, we could better stratify patients for gene-targeted therapies. For example, individuals harboring high-risk variants may benefit from earlier or more aggressive intervention, or may be ideal candidates for enrollment in clinical trials testing variant-specific treatments.”

  • The last paragraph of the introduction contains information that concerns the methodology (lines 79-87).

We agree with the reviewer and have moved the following text to section 2.2:

“gnomAD is a publicly available resource that aggregates and harmonises exome and genome sequencing data from 730,947 exomes and 76,215 whole genomes from over 800,000 individuals, including ~138,000 individuals from non-European groups. It serves as a valuable resource for studying genetic variation in the general population, as it integrates data from both disease-specific studies and general population cohorts while excluding individuals with severe pediatric diseases.”

  • The last paragraph of the Materials and Methods section is a copy of the Genes editors' instructions for preparing manuscripts - it should be removed.

We thank the reviewer for this observation and apologise for the overlook, we have taken out the instructions paragraph from the manuscript.

  • Figure 1: the scale of the Y-axis should be wider, because some values are higher than indicated on the Y-axis (applies to both A and B).

We thank the reviewer for their comment. Upon careful examination, we identified an inconsistency in the figure legend, which initially described Panel A as displaying all variants found in ClinVar and OMIM (1,849 variants). However, Panel A actually depicts only the variants present in the gnomAD dataset (368 variants), leading to a discrepancy between the legend and both the total variant count and the Y-axis scale.

Panel B, on the other hand, displays only significant variants, ranking  them based on the population in which each variant is most frequent. Each of the 81 variants is assigned to a single population, without overlap, as each variant can only be assigned to a single population.

Additionally, following the recommendation of Reviewer 2, we have generated a new figure to replace Figure 1A. As a result, we now have three figures: Figure 1 (replacing 1A), Figure 2 (formerly 1B), and Figure 3 (formerly Figure 2).

To enhance clarity, we have also revised the figure legends to better reflect what is being depicted. The revised legends now read:

“Fig 1. Number of pathogenic variants reported per ancestry and overlap between populations

The Y-axis lists genes included in this study, ordered by the total number of pathogenic/likely pathogenic (P/LP) variants reported. The X-axis shows the total number of P/LP variants found in ClinVar/OMIM per ancestry, with colored bars representing variants found in the gnomAD dataset per population. Each population includes both exclusive and shared variants. To the left of each bar, a half-matrix indicates the overlap of variants between populations. Each number in the matrix represents the count of overlapping variants between the population associated with the corresponding colored bar and the population listed in each column. Panel A lists genes with more than 18 variants reported (population with highest count). Panel B lists the genes with less than 18 variants reported (population with highest count). AFR: African/African-American; AMR: Admixed American; ASJ: Ashkenazi Jewish; EAS: East Asian; FIN: Finnish; MID: Middle Eastern; SAS: South Asian; NFE: Non-Finnish European.”

“Fig 2. Number of significant pathogenic variants most frequent across ancestry.

The X-axis lists the genes included in this study in alphabetical order. The plot shows the number of significant variants assigned to each population, where each variant was categorized based on the population in which it had the highest allele frequency. In this panel, the Y-axis represents the number of variants accumulated by each population for every gene. AFR: African/African-American; AMR: Admixed American; ASJ: Ashkenazi Jewish; EAS: East Asian; FIN: Finnish; MID: Middle Eastern; SAS: South Asian; NFE: Non-Finnish European.”

“Fig 3. …”

  • It would be worth adding a table containing data on previously identified and new genes, the variants of which are associated with PD.

We appreciate the reviewer's suggestion. To address this, we have included Supplementary Table 1, which lists all genes previously associated with Mendelian forms of Parkinson’s Disease, along with their primary phenotypic associations as reported in OMIM. Additionally, we have compiled Supplementary Table 2, which provides details on the specific comparable variants analysed in our study, including their associated conditions and genotypes based on ClinVar and OMIM.

Our study was designed specifically to compare allele frequencies of previously identified pathogenic and likely pathogenic variants within established Mendelian PD genes. As such, identifying new genes was beyond the scope of our analysis. We hope this clarification is helpful and appreciate the reviewer’s thoughtful input.

  • Unfortunately, the link provided as a reference to the source of supporting information www.mdpi.com/xxx/s1 does not work - on the page that opens at this address, the message Error 404 - File not found is displayed

To our understanding, the link mentioned works as a placeholder for the site that will be allocated for the supplementary material once the review process is over and the paper is published.

Reviewer 2 Report

Comments and Suggestions for Authors

The authors do an analysis where they examine genes associated with Mendelian forms of Parkinson's Disease and examine differences in variant selection, allele frequency, geographic distribution, and more. The data used is from an open access data resource. Results show significant differences that are discussed and interpreted. 

Overall the paper is interesting and provides a standard piece-wise methodological framework. However, there is concern that the integrative methodology, as presented, may be resulting in excessive false positives and/or excessive false negatives.  The thresholds used for identifying statistical significance and/or selection for percentiles were not made clear and/or were not well supported with explanation.  The analysis (Figure 1) to compare geographic distributions was not well supported analytically or pictorially. Improved visualization to better examine intersections/overlap and/or differences for number of variants would have been more helpful. This should also include visual post-hoc analysis of significant differences.

As is, the authors tend to find the already obvious genes or find genes who importance could be overly tied to subjective thresholding.  In order to better support the conclusions, the authors will need to carefully illustrate their methodology to minimize even perceived possible inflation of Type 1/2 errors based on thresholds or multiple rounds of selection. Minimally, an overall data analysis pipeline figure with the justified thresholds for selection at each stage is necessary.

All in all, the study has a good foundation, but more improvements are necessary to clarify the methodology and to improve visualization that clearly supports the stated conclusions.

MINOR:

Authors left the MDPI template text paragraph instructions for Methods under paragraph 1 of their own Methods. Please remove.

The authors state "code can be obtained upon reasonable request". It would appear that for a study that used entirely open access datasets that providing the code in a GitHub repo or something similar should be standard.

Author Response

Reviewer 2

The authors do an analysis where they examine genes associated with Mendelian forms of Parkinson's Disease and examine differences in variant selection, allele frequency, geographic distribution, and more. The data used is from an open access data resource. Results show significant differences that are discussed and interpreted.

We thank the reviewer for taking the time to read and review our study. Thanks to the comments and observations raised by the reviewer, we had the opportunity to strengthen the robustness of the study and improve the work presented. Below, we respond to all observations.

  • Overall the paper is interesting and provides a standard piece-wise methodological framework. However, there is concern that the integrative methodology, as presented, may be resulting in excessive false positives and/or excessive false negatives. The thresholds used for identifying statistical significance and/or selection for percentiles were not made clear and/or were not well supported with explanation.

We thank the reviewer for raising this important question and for the opportunity to clarify our methodology. To ensure transparency regarding our statistical approach and thresholding criteria, we have revised Section 2.2 ("Testing for allele frequency differences between populations") to provide a more detailed explanation of the allele frequency comparisons and statistical significance thresholds. The revised paragraph now reads:

“To assess differences in allele frequencies, we performed Fisher’s exact tests for all possible pairwise comparisons between populations, generating contingency tables for each variant in each comparison using minor and major allele counts from the populations being compared. Major allele counts were calculated by subtracting the minor allele count from the ‘Allele Number (AN)’ in gnomAD’s VCF files, which represents the total number of alleles analysed for a given SNP in a population. To ensure meaningful comparisons, we excluded cases where a population had a minor allele count of zero or where allele count data were unavailable.

To account for false positives, we applied a false discovery rate (FDR) correction using the Benjamini-Hochberg procedure. Considering that pairwise population comparisons are independent of one another, we performed FDR correction separately for each set of comparisons. However, within each pairwise comparison set, variants were grouped by gene, as variants in the same gene are not entirely independent due to linkage disequilibrium. Thus, FDR correction was applied to adjust the P-values of all variants in every gene, independently across pairwise comparison sets. Only variants with an FDR-adjusted P-value < 0.05 were considered statistically significant. (Table S3)

The revised portion of the methods reflects with more accuracy the measures taken in order to minimise type one error by setting a conservative approach of false discovery rate correction on all P values obtained directly from the Fisher Tests performed. Additionally, we revised section 3.2 of the results section to clarify that only significant variants were considered for reporting the top 10th percentile of global allele frequency according to gnomAD. The revised portion of the text now reads:

“Significant variants in the top 10th percentile of global allele frequency were identified in the HTRA2, PRKN, GBA1, LRRK2, GIGYF2 and EIF4G1 genes.”

  • The analysis (Figure 1) to compare geographic distributions was not well supported analytically or pictorially. Improved visualization to better examine intersections/overlap and/or differences for number of variants would have been more helpful. This should also include visual post-hoc analysis of significant differences.

We thank the reviewer for their valuable feedback and for highlighting these important points. In response, we have replaced Panel A of Figure 1 with a new plot that illustrates the overlap of variants between populations across genes. Additionally, we have retained Panel B as a post-hoc analysis, focusing only on significant variants. To improve clarity and consistency, the revised figures are now designed as follows:

Figure 1 (replacement for the original Figure 1A), Figure 2 (formerly Figure 1B) and Figure 3 (formerly Figure 2).

Revised figure legends now read :

“Fig 1. Number of pathogenic variants reported per ancestry and overlap between populations

The Y-axis lists genes included in this study, ordered by the total number of pathogenic/likely pathogenic (P/LP) variants reported. The X-axis shows the total number of P/LP variants found in ClinVar/OMIM per ancestry, with colored bars representing variants found in the gnomAD dataset per population. Each population includes both exclusive and shared variants. To the left of each bar, a half-matrix indicates the overlap of variants between populations. Each number in the matrix represents the count of overlapping variants between the population associated with the corresponding colored bar and the population listed in each column. Panel A lists genes with more than 18 variants reported (population with highest count). Panel B lists the genes with less than 18 variants reported (population with highest count). AFR: African/African-American; AMR: Admixed American; ASJ: Ashkenazi Jewish; EAS: East Asian; FIN: Finnish; MID: Middle Eastern; SAS: South Asian; NFE: Non-Finnish European.”

“Fig 2. Number of significant pathogenic variants most frequent across ancestry.

The X-axis lists the genes included in this study in alphabetical order. The plot shows the number of significant variants assigned to each population, where each variant was categorized based on the population in which it had the highest allele frequency. In this panel, the Y-axis represents the number of variants accumulated by each population for every gene. AFR: African/African-American; AMR: Admixed American; ASJ: Ashkenazi Jewish; EAS: East Asian; FIN: Finnish; MID: Middle Eastern; SAS: South Asian; NFE: Non-Finnish European.”

  • As is, the authors tend to find the already obvious genes or find genes who importance could be overly tied to subjective thresholding. In order to better support the conclusions, the authors will need to carefully illustrate their methodology to minimize even perceived possible inflation of Type 1/2 errors based on thresholds or multiple rounds of selection. Minimally, an overall data analysis pipeline figure with the justified thresholds for selection at each stage is necessary.

 All in all, the study has a good foundation, but more improvements are necessary to clarify the methodology and to improve visualization that clearly supports the stated conclusions.

We thank the reviewer for their insightful recommendation. To address this concern, we have created Supplementary Figure 1, which clearly outlines the analysis pipeline used in this study to assess differences in allele frequency between populations. This figure details each step of the selection process, including the number of variants retained at each stage and the specific thresholds applied, ensuring transparency in our methodology. Additionally, we have revised our visualisation strategies and restructured all figures to enhance clarity and better support our conclusions. We appreciate the reviewer’s feedback, which has helped us improve the rigour and presentation of our study.

MINOR:

  • Authors left the MDPI template text paragraph instructions for Methods under paragraph 1 of their own Methods. Please remove.

We thank the reviewer for this observation and apologise for the overlook, we have taken out the instructions paragraph from the manuscript.

  • The authors state "code can be obtained upon reasonable request". It would appear that for a study that used entirely open access datasets that providing the code in a GitHub repo or something similar should be standard.

We agree with the reviewer and have made available a git repository with all the code used to conduct the analysis. We have included the link in the data availability statement.

This will be ideally supplementary, I would say

Round 2

Reviewer 2 Report

Comments and Suggestions for Authors

Authors have addressed major points.